# Peer review of "TiO2 Self-Assembled, Thin-Walled Nanotube Arrays for Photonic Applications"

_materials, 2019, doi:10.3390/ma12081332_

Reviewer 1 Report

In this paper, the authors introduced a  photonic crystal structures with respect to reduction of front surface reflection, achievable field enhancement, and photonic bands. With the Rigorous Coupled Wave Analysis (RCWA), we study the optical response of  photonic crystals made of thin-walled nanotubes relative to the bare Ti foil substrate, including under additional charge carrier doping. The idea behind this is interesting. However, I still have quite a number of concerns in this manuscript. There are times where there are not enough data to support the conclusions of the author. Please see some of the major concerns below.

1.The information for the PC structure with the nanotube is not enough. The authors should give much more information about this. So the readers can get its reproducibility, for an example can authors show the 3D full image of the device?  Where is the pitch?  And diameter of the PC? 

2.  The authors should give much more information about the novelty of this paper, especially the effect of using this structure, which applications can be used this device?

3. The fabrication tolerance analysis, which can offer a good guide for the fabrication requirement, and the key parameters, need to be added in the results section.

4. More references need to be included in the introduction part to understand the applications of using PC structures and polymer material

"Prospects for diode pumped alkali atom      based hollow core photonic crystal fiber lasers",Optics Letters,      39(16), 2014 (4655-4658).

2.     "A visible light RGB wavelength demultiplex based on silicon-nitride multicore PCF", Optics & Laser Technology, 111, 2019 (411-416)

3.     "An eight-channel C-band demux based on multicore photonic crystal fiber"

         Nanomaterials, 8(10), 2018, 845 (10 pages)

Author Response

response letter attached

Reviewer 2 Report

The article concerns the modeling of TiO2 nanotubes for selected optical properties (behavior analogous to photonic crystals, plasmon generation, etc.). As such, the article fits the profile of the journal, although the uniqueness is limited by the only computational nature of the work (no verification). The structure of the article, the way of describing scientific papers and the presentation of applications are confusing and significantly hamper the use of the article and the assessment of the usefulness of the material. Detailed remarks below.

Line 50 "The calculations were performed using RI data for TiO2 from Ref. [25] and Ti from Ref. [26]". It woule be easier for the reader  if the Author provide values and references not only references.

The author convincingly presented in the introduction the motivation to undertake work, but she illegally described their scope. Does the work concern only modeling or also experimental verification? If the first one, how has the accuracy and correctness of the calculations been verified?

It is very difficult to find information what structures were modeled, real or ideal. Can the author describe the object of modeling in a separate subsection?

Tools were not described. I do not assume that the Author conducted calculations manually.

Line 147. Doping by what? Simulated materials/structures should be clearly described. Table?

Line 164-165 "Higher doping levels yield plasmon-like behavior and make these structures interesting for sensing and spectroscopy applications" - too general - what applications? Compare with existing plasmon-based solutions (show advantages).

Conclusions are definitely too laconic and categorical. There is no justification for the detailed conclusions presented.

Author Response

detailed reply letter attached

Reviewer 3 Report

The author show a possibility of the local field enhancement at specific wavelengths in photonic crystal system consisting of TiO2 hollow nanotubes. It is shown that wall thickness of the nanotube  or doping level occuring during the growth process from Ti foil can have strong effect on the optical properties of the nanotube array. In particular, it can maximize the local field enhancement or yield plasmon like behavior. The author also demonstrate that the TiO2 nanotube arrays demonstrate properties of photonic crystals. I think the paper can be published in Materials.  

Author Response

The author thanks the reviewer for their positive assessment of the submitted manuscript.

Round  2

Reviewer 1 Report

The new version can be published.

Reviewer 2 Report

I am satisfied with the Author response.